# Effects of Eugenol on Water Quality and the Metabolism and Antioxidant Capacity of Juvenile Greater Amberjack (*Seriola dumerili*) under Simulated Transport Conditions

**DOI:** 10.3390/ani12202880

**Published:** 2022-10-21

**Authors:** Yuhang He, Zhengyi Fu, Shiming Dai, Gang Yu, Yunfeng Guo, Zhenhua Ma

**Affiliations:** 1Sanya Tropical Fisheries Research Institute, Sanya 572018, China; 2Hainan Provincial Key Laboratory of Efficient Utilization and Processing of Marine Fishery Resources, Sanya 572018, China; 3Fisheries Management and Law Enforcement Service Centre, Ministry of Agriculture and Rural Affairs, Shanghai 200092, China; 4Tropical Aquaculture Research and Development Center, South China Sea Fisheries Research Institute, Chinese Academy of Fishery Sciences, Sanya 572018, China

**Keywords:** Eugenol, liver, gill, metabolism capacity, antioxidant capacity, water quality

## Abstract

**Simple Summary:**

The transportation of live fish is a very important part of the aquaculture system. The health status of fish can be affected in the closed-container transportation system. It has become very common to relieve the stress of fish in transport by adding anesthetics. With the food safety of fish having gained widespread attention, the addition of plant extracts as an alternative to anesthetics has become an important research topic. The objective of this study was to evaluate the effects of eugenol addition during simulated transport on water quality, and the metabolism and antioxidant capacity of liver and gills in the greater amberjack (*Seriola dumerili*). It was found that the addition of eugenol under simulated transport conditions had positive effects on water quality, and on liver and gill metabolism and antioxidant capacity in the greater amberjack. This study contributes to the healthy culture of the greater amberjack.

**Abstract:**

This study investigated the effects of added eugenol on water quality and the metabolism and antioxidant capacity of the liver and gills of the greater amberjack (*Seriola dumerili*) during simulated transport. The juvenile fish (10.34 ± 1.33 g) were transported in sealed plastic bags containing different eugenol concentrations at a density of 24.79 kg/m^3^ for 8 h. The different eugenol concentrations were divided into five groups: 0 μL/mL (control group), 0.0125 μL/mL, 0.025 μL/mL, 0.0375 μL/mL, and 0.05 μL/mL, with three replicates of each. The results showed that 0.05 μL/mL of eugenol could significantly increase dissolved oxygen, but 0.025 μL/mL–0.0375 μL/mL resulted in a significant decrease in dissolved oxygen and significant increases in NH_4_^+^-N and NO_2_^−^-N. It was found that 0.05 μL/mL of eugenol caused significant up-regulation of the relative expression of CPT-1 in the liver, significant down-regulation of the relative expression of FAS and PK in the liver and gills, a significant increase in glycogen concentration, and a significant decrease in glucose concentration. This suggests that 0.05 μL/mL of eugenol could reduce the metabolic capacity of fish. In addition, 0.05 μL/mL of eugenol caused significant up-regulation of the relative expression of CAT and a significant decrease of MDA concentration in the liver. Meanwhile, the gills showed significant up-regulation of CAT relative expression, significant down-regulation of Keap1 relative expression, and a significant increase in GSH activity, resulting in a significant increase in MDA concentration when the concentration of eugenol reached or exceeded 0.025 μL/mL. This suggests that 0.05 μL/mL eugenol could improve the antioxidant capacity of fish and lipid peroxidation levels in the gills. In conclusion, the addition of 0.05 μL/mL eugenol could improve water quality, and the metabolic and antioxidant capacities of liver and gills, but it could also increase lipid peroxidation levels in the gills under transport conditions.

## 1. Introduction

FAO (Food and Agriculture Organization of the United Nations) data show that marine fishery resources are continuously declining, but people’s consumption of aquatic animals has increased at an average annual rate of 3.1% in the last two decades [1]. The rapid development of aquaculture provides a valuable source of protein for the growing global human demand. The transportation of living fish plays an important role in global aquaculture production, and usually occurs in the contexts of transporting juvenile fish from hatchery fry to grow-out ponds, transferring adult fish to fish markets, and releasing juveniles for stocking programs or to aquaria [2]. However, aquatic animals can be affected by transport stress in the closed-container transportation system [3]. Transport stressors include high stocking density, noise, vibration, and deterioration of water quality, which can lead to systemic disruptions in aquatic animal physiology and to high mortality [4,5]. The balance between the reactive oxygen species (ROS) and antioxidant capacity of the fish organism can be broken when changes in the external environment occur during transport [6]. A large amount of ROS accumulates and leads to oxidative stress, which causes damage to normal cells and tissues, and even leads to lipid peroxidation [7]. Furthermore, the non-specific immune system of fish produces a range of physiological, biochemical, and immunological changes in response to oxidative stress. The antioxidant systems of fish are composed of non-enzymatic compounds, such as SOD, CAT, GSH, and GR. These antioxidant compounds and enzymes convert and decompose superoxide anions and hydrogen peroxide to protect cells from damage caused by the accumulation of (ROS) [7]. ROS can also activate transcription factors to control antioxidant enzymes, such as Nrf2/Keap1 which binds to the cytosolic chaperone Keap1 to maintain relative inhibitory activity under physiological conditions [8,9]. When ROS accumulates in excess, Nrf2 activates a series of downstream antioxidant enzymes, such as SOD and CAT [10]. These antioxidant indicators are closely linked to the entire immune defense system.

The greater amberjack (*Seriola dumerili*) has attracted more and more attention from the global aquaculture industry due to its tender flesh, high nutritional value, and rapid growth [11,12]. The farming of the greater amberjack relies heavily on the transportation of live fish, because its juveniles are usually captured in the wild. However, to the best of our knowledge, little information is available about reducing the transportation stress of the greater amberjack. Currently, the addition of anesthetics has been widely employed to reduce stress during transport, such as MS-222, benzocaine hydrochloride, 2-phenoxyethanol, and lidocaine hydrochloride [13,14]. Anesthetics reduce the sensitivity of fish to stressors by inhibiting the central nervous system, and reduce metabolism and oxygen uptake, thereby reducing CO_2_ and ammonia production, maximizing the welfare of numerous fish species [15,16,17]. However, the edible safety of fish treated with anesthetics has aroused great concern, and questions remain about the efficiency and safety of these synthetic anesthetics as residues in human food, leading researchers to seek easily available, low-cost, and safe anesthetics [18,19]. Plant extracts or essential oils may be viable alternatives to fish anesthetics, such as basil *Ocimum basilicum* essential oil, rubber seeds (*Hevea brasiliensis*), and tuba root (*Derris elliptica*) [20,21,22,23].

Among many natural agents, clove oil is the most commonly used and has a reputation as the most effective fish anesthetic [24,25,26]. Eugenol [2-methoxy-4(2-propenyl) phenol] is the main component of clove oil, which is extracted from the distillation of leaves, flowers, and stems of the clove tree *Syzygium aromaticum* [27]. It is considered a suitable anesthetic because of its high efficacy, low price, lack of withdrawal period, and lack of negative effects for fish and humans [28,29]. Eugenol has been officially approved as an anesthetic for food fish in many countries, including Japan, New Zealand, and some southeast Asian countries [30]. Previous studies have reported that eugenol significantly reduces transportation stress in *Rhamdia quelen* and *Lophiosilurus alexandri* [31,32]. While most of the studies on reducing the transport stress of aquatic animals have focused on physiological and immune indicators of serum, the health of the gills and liver after transportation is also very important [31,32]. Owing to the location of the gills between the external and internal environments, they play crucial roles in gas transfer, acid-base balance, and ionic regulation [33]. Fish liver is the main organ of detoxification, and is involved in numerous functions including metabolism and immune system processes [34]. The condition of these organs determines the growth, survival, and health of the fish. Therefore, it is necessary to investigate the effects of eugenol on the gills and liver of the greater amberjack after transport. We investigated the effects of eugenol on juvenile greater amberjack under simulated transport conditions, through changes in water quality and metabolism-related and antioxidant-related indicators of the liver and gills. This study aimed to investigate whether eugenol could alleviate the stress response of the greater amberjack, and to determine a reasonable concentration of eugenol. The results provide information for the application of plant extracts as anesthetics in aquaculture. 

## 2. Material and Methods

### 2.1. Ethics Statement

This study was carried out in strict accordance with the recommendations of the Animal Welfare Committee of the Chinese Academy of Fishery Science. No protected species were used during the experiment.

### 2.2. Experiment Design and Fish

The eugenol used in the experiment was supplied by Shanghai Yuanye Bio-Technology Co., Ltd. (98.5% purity, Shanghai, China). We referenced the eugenol concentrations from previous studies on yellow perch *Perca flavescens* (Mitchill) and Nile tilapia *Oreochromis niloticus* L [35]. The tests included five eugenol treatment groups with three replicates for each eugenol treatment group, and D0 was the control group without eugenol. Different doses of eugenol stock solution (0.5 mL, 1 mL, 1.5 mL, and 2.0 mL, respectively) were added to the water to develop different eugenol treatment groups (D25: 0.0125 μL/mL, D50: 0.025 μL/mL, D75: 0.0375 μL/mL, D100: 0.05 μL/mL). 

The wild-caught juvenile amberjack used in the experiment were provided by Tropical Fisheries Research and Development Center, South China Sea Fisheries Research Institute, Chinese Academy of Fishery Science (Lingshui, Hainan, China). The juveniles were acclimatized for 15 days, during which they were fed commercial feed (49.7% crude protein, 12.7% crude lipid) twice daily while kept in an 800 L circulating mariculture tank. Water temperature, dissolved oxygen, pH, and salinity were maintained at 29 °C, above 6.0 mg/L, 7.87, and 35 ppt, respectively (Hach HQ40D, USA). Juvenile fish of similar size (weight: 10.34 ± 1.33 g, length: 19.83 ± 0.02 cm) fasted for 24 h were selected and randomly allocated to 15 sealed plastic bags containing 40 L seawater at density of 24.79 kg/m^3^. The bags were subsequently inflated with pure oxygen, sealed with a rubber band, and then placed in sealed foam boxes containing ice bottles (500 mL) for cooling. The duration of transport simulation was 8 h, during which the foam boxes were shaken 5 times per hour for 1 min each time to simulate the turbulence of transportation. Water, gill, and liver samples were taken immediately after the simulated transport. The schematic design of the experiment is presented in Figure 1.

### 2.3. Sampling

At the end of the transport experiment, 100 mL samples of water were gathered from each bag and used for water quality analysis. Five fish (*n* = 5) were randomly selected from each bag for tail vein extraction of blood, and gill and liver tissue sampling. The gill and liver samples were rapidly frozen by liquid nitrogen and stored at −80 °C for analysis of antioxidant capacity and metabolic capacity. Blood samples were placed at 4 °C for 3 h and then centrifuged in a cooled centrifuge (4 °C, 3000 rpm), and the separated serum was stored at −80 °C until glucose concentration analysis.

### 2.4. Measurement of Water Quality

The pH and dissolved oxygen (DO) of each water sample were measured by a portable water quality monitor (Hach HQ40D, USA), and each water sample was tested for NH_4_^+^-N and NO_2_^−^-N according to standard methods [36].

### 2.5. Assay of Metabolic and Antioxidant-Related Enzymes 

Liver and gill tissues were homogenized in ice-cold saline (0.86%) at a proportion of 1:9 (*W*/*V*). After centrifugation (2500 rpm, 10 min, 4 °C), the glutathione activity (GSH, A006-2-1) and the concentrations of malondialdehyde (MDA), glycogen, and glucose were analyzed using commercially available kits according to the instructions provided (Nanjing Jiancheng Biotech. Co., Nanjing, China). The protein concentration of extracts was measured using the Bradford method [37].

### 2.6. Relative Expression of Metabolic and Antioxidant-Related Genes

The harvested frozen liver and gill tissues were homogenized on ice by a hand-held homogenizer (Greenprima Instruments Co., Ltd., UK) in 1 mL of trizol (Invitrogen, Thermo Fisher Scientific Co., Ltd., Shanghai, China), and the RNA was isolated in a chloroform layer and precipitated with isopropanol. Then, the quantity of isolated RNA was determined by measuring its absorbance at 260 and 280 nm using a Nano-300 Micro Spectrophotometer (Hangzhou Allsheng Instruments Co., Ltd., China), and the quality of total RNA was detected by electrophoresis with 1.0% agarose gel. Qualified total RNA was reverse transcribed according to the manufacturer’s instructions with a One-Step gDNA Removal and cDNA Synthesis SuperMix (EasyScript, Beijing TransGen Biotech., Ltd., Beijing, China), and the first-strand cDNA was synthesized accordingly. Real-time quantitative PCR analysis was conducted using a real-time PCR system (Q1000, Hangzhou Longji Scientific Instruments Co., Ltd., Hangzhou, China). The antioxidant-related genes were selected from the *Seriola dumerili* NCBI database (http://www.ncbi.nlm.nih.gov, accessed on 10 March 2022) for analysis by qPCR. The running conditions were as follows: initial denaturation at 95 °C for 15 min, 40 cycles of 95 °C for 10 s, 60 °C for 20 s, and 72 °C for 30 s. All PCR amplifications were performed in triplicate, the reactions included two measures of RealUniversal PreMix (10 μL), each with PCR primers (0.6 μL, 10 µmol/L), template cDNA (2 μL), and RNase-free dd H_2_O, with a final volume of 20 μL. Our team screened the housekeeping genes before the experiment. The results showed CT ranges of 20.21 ± 1.77 for β-actin and 18.81 ± 1.15 for EF1-α, with M values between 1 and 1.5. Therefore, EF1 α and β-Actin were used as the reference genes. The melting curve analysis was used to check and verify the specificity and purity of each PCR at the end of every PCR process. The 10-fold serial dilution of each cDNA primer pair was performed in order to establish a standard curve. The efficiency of PCRs ranged from 90% to 110% with Pearson’s coefficients of determination (*R*^2^) > 0.97. The primers used for the analysis are shown in Table 1. We determined the relative expression of carnitine palmitoyltransferase-1 (CPT-1), fatty acid synthetase (FAS), pyruvate kinases (PK), catalase (CAT), manganese-superoxide dismutase (Mn-SOD), glutathione reductase (GR), and Keap1 in liver and gill tissue, with reference to previous studies that selected antioxidant-related and metabolism-related genes [38,39,40].

### 2.7. Statistical Analysis

The expression levels of antioxidant-related genes were normalized based on the level of the housekeeping genes (EF1-α and β-Actin). Gene expression was analyzed based on the 2^−∆∆Ct^ method [41]. All data are presented as the standard deviation (mean ± SD). One-way analysis of variance (ANOVA) was used to test the effect of adding eugenol. All data were tested for normal distribution and homogeneity of variance. Where significance was detected, LSD-test multiple comparisons were used to compare means between groups. All statistical analysis was performed with SPSS 23.0 (IBM, Armonk, NY, USA) software. Significant difference was defined as *p* < 0.05.

## 3. Results

### 3.1. Water Quality after Simulated Transport Conditions

DO, pH, NH_4_^+^-N, and NO_2_^−^-N levels in water with different eugenol concentrations after transportation simulation are shown in Table 2. There was no significant difference between the pH values of the groups. Significantly decreased (*p* < 0.05) DO was found in the D50 and D75 groups, while a significantly increased (*p* < 0.05) DO was found in the D100 group compared with the control group. Significantly increased (*p* < 0.05) NH_4_^+^-N was found only in the D50 and D75 groups. Significantly increased (*p* < 0.05) NO_2_^−^-N was found in the D50 and D75 group compared with the control group.

### 3.2. Metabolic Capacity after Simulated Transport Conditions 

Metabolism-related genes of the liver and gill tissue of juvenile greater amberjack after simulated transport under different eugenol concentrations are shown in Figure 2. In the liver, significantly up-regulated (*p* < 0.05) relative expression of CPT-1 was found in the D75 and D100 groups, with the maximum value was found in the D75 group. Significantly down-regulated (*p* < 0.05) relative expression of FAS was found in the D25, D50, and D100 groups, with the minimum value found in the D25 group, and significantly up-regulated (*p* < 0.05) relative expression found only in the D75 group. Significantly down-regulated (*p* < 0.05) relative expression of PK was found in all eugenol-containing groups. In the gills, significantly down-regulated (*p* < 0.05) CPT-1 relative expression was found in the D25, D50, and D100 groups, with the minimum value found in the D50 group. Significantly up-regulated (*p* < 0.05) relative expression of FAS was found in the D50 and D75 groups, while significantly down-regulated (*p* < 0.05) relative expression was found only in the D25 group; the maximum value was found in the D75 group. Significantly up-regulated (*p* < 0.05) relative expression of PK was found only in the D25 group, while down-regulated relative expression was found only in the D75 group. The linear regression equations for the relative expression of CPT-1, FAS and PK in liver and gill tissues are shown in Appendix A.

Concentrations of serum glucose and liver glycogen are shown in Figure 3. Significantly higher (*p* < 0.05) liver glycogen concentrations were found in all eugenol-containing groups, with the maximum value found in the D75 group. Significantly higher serum glucose concentrations were found in the D25 and D50 groups, while significantly lower (*p* < 0.05) concentrations were found only in the D100 group. The linear regression equations of eugenol with liver glycogen and serum glucose concentrations are shown in Appendix A.

### 3.3. Antioxidant Capacity after Simulated Transport Conditions

Antioxidant-related indicators from the liver tissue of juvenile greater amberjack after simulated transport conditions with different eugenol concentrations are shown in Figure 4. The relative expression of CAT was significantly up-regulated (*p* < 0.05) in all eugenol-containing groups compared with the control group, with the maximum value found in the D100 group. Significantly up-regulated (*p* < 0.05) relative expression of Mn-SOD was found only in the D50 group. The relative expression of GR first decreased and then increased with the increase of eugenol level, and the minimum value was found in the D50 group, with no significant difference (*p* > 0.05) between each of the eugenol-containing groups and the control group. Significantly down-regulated (*p* < 0.05) relative expression of Keap1 was found in the D25 and D50 groups, while significantly up-regulated (*p* < 0.05) relative expression of Keap1 was found in the D75 and D100 groups, with the maximum value found in the D75 group and the minimum value found in the D25 group. The activity of GSH in the liver was significantly decreased (*p* < 0.05) in all eugenol-containing groups compared with the control group, with the minimum value found in the D75 group. Significantly lower (*p* < 0.05) MDA concentration in the liver was found in the D25, D50, and D100 groups. The linear regression equations of eugenol on antioxidant-related genes (and antioxidant-related enzymes) in the liver are shown in Appendix A. 

Antioxidant-related indicators from gill tissue of juvenile greater amberjack after simulated transportation with different eugenol concentrations are shown in Figure 5. The relative expression of CAT was significantly up-regulated (*p* < 0.05) in all eugenol-containing groups compared with the control group, with the maximum value found in the D75 group. In contrast, significantly down-regulated (*p* < 0.05) relative expression of Mn-SOD was found in all eugenol-containing groups compared with the control group, with the minimum value found in the D50 group. Significantly up-regulated (*p* < 0.05) relative expression of GR was found only in the D50 group. Significantly down-regulated (*p* < 0.05) relative expression of Keap1 was found in the D50, D75, and D100 groups, with the minimum value found in the D50 group. GSH activity in the gills was positively associated with eugenol levels, with significantly increased (*p* < 0.05) activity found in all eugenol-containing groups. Compared with the control group, significantly increased (*p* < 0.05) MDA concentration was found when the eugenol content reached or exceeded 0.025 μL/mL, with the maximum value in the D50 group. The linear regression equations of eugenol on antioxidant-related genes and antioxidant-related enzymes in the gills are shown in Appendix A.

## 4. Discussion

### 4.1. The Effects of Eugenol on Liver and Gill Metabolic Capacity after Simulated Transport

The addition of appropriate eugenol levels could reduce the metabolism of liver and gills under live transport conditions. These results were similar to previous studies, such as those reporting significantly lower metabolic rates with eugenol in yellow perch and Nile tilapia [35]. In this experiment, eugenol modulated the relative expression of FAS, CPT-1, and PK in the liver and gills. FAS is a key enzyme in fat synthesis, facilitating the de novo synthesis of fatty acids by catalyzing the synthesis of fatty acids from acetyl-CoA and malonyl-CoA [42]. CPT-1 plays a crucial role in the β-oxidative catabolism of fatty acids for energy supply, which can maintain the balance of blood glucose and energy supply when the body or tissues are deficient in energy [43]. PK is an important enzyme in the glycolytic pathway, catalyzing the conversion of phosphoenolpyruvate to enolpyruvic acid and the production of ATP [44]. Eugenol at 0.05 μL/mL could up-regulate the relative expression of CPT-1 and down-regulate the relative expression of FAS and PK in liver tissue, which indicates that eugenol could reduce the synthesis of glycolysis and lipids while increasing lipid catabolism. When the eugenol concentration was increased to 0.05 μL/mL, the relative expression of PK in gill tissue was significantly down-regulated and DO was significantly increased, which suggests that 0.05 μL/mL of eugenol reduces energy requirements in the gills by decreasing glycolysis. 

The addition of unsuitable concentrations of eugenol can increase transportation stress in the gills. Studies have shown that when stress is present, it is usually accompanied by a decrease in CPT-1 and an increase in FAS, consistent with the results for the D50 and D75 groups [45]. We believe this may be caused by deteriorating water quality. The water quality results for D50 and D75 showed significant decreases in DO and significant increases in NH_4_^+^-N. The water environment is the main medium for fish to carry out gas exchange and ion exchange, and for purification of metabolites. The water environment during transportation is characterized by small water volume and high density of fish, so the water quality during live transport is very susceptible to significant changes caused by the influence of external factors. Inappropriate eugenol concentrations could increase the gills’ metabolic capacity by down-regulating the relative expression of CPT-1 and up-regulating the relative expression of FAS and PK. Increased metabolic capacity including ammonia excretion rate and respiration may negatively affect water quality and lead to transport stress in the gills. 

An appropriate quantity of eugenol can relieve transport stress in the liver and gills by reducing metabolic capacity. The experimental results showed a significant increase in liver glycogen concentration and a significant decrease in serum glucose when eugenol was added at a concentration of 0.05 μL/mL. This may indicate that eugenol reduces the metabolic energy demands of the greater amberjack during transport, meaning that it requires less energy to cope with external stress. Glucose is an important source of energy that ensures the normal functioning of various tissues. Glycogen serves as a form of energy storage in animals and is stored primarily in the cells of the liver. The body maintains a relatively constant glucose concentration through the synthesis and catabolism of liver glycogen [46]. Previous studies have shown that aquatic animals respond to the stress of low temperatures and transport by breaking down liver glycogen to maintain the body’s energy requirements [47,48]. Eugenol has also been found by previous studies to relieve transportation stress, with significantly reduced serum levels of COR (cortisol) and GOT (glutamic oxaloacetic transaminase) found in hybrid amazon catfish (*Pseu-doplatystoma reticulatum* × *Leiarius marmoratus*) anesthetized by eugenol [49]. Furthermore, the results of liver glycogen, blood glucose, and metabolism-related gene tests showed reduced glycolytic capacity in the liver and gills. Glycolytic capacity is usually positively correlated with stress, which is a relatively rapid and intense biochemical process that usually occurs during intense exercise [50]. This may also suggest that eugenol could reduce stress in fish during transport by decreasing their motility. Therefore, the addition of 0.05 μL/mL eugenol was found to be the optimum concentration with regard to the metabolic capacities of the liver and gills in greater amberjack under transportation conditions.

### 4.2. The Effects of Eugenol on Liver and Gill Antioxidant Capacity after Simulated Transport

The recommended concentration of eugenol during simulated transport is less than 0.025 μL/mL when considering the antioxidant capacity of the greater amberjack. The addition of an appropriate concentration of eugenol could improve the antioxidant capacity of the liver of the greater amberjack under transportation conditions. The results showed that relative expressions of CAT and Mn-SOD were significantly up-regulated in the livers of the group containing the optimal eugenol concentration, while the relative expression of Keap1 was significantly down-regulated. However, positive modulation of antioxidant genes cannot directly represent enhanced antioxidant capacity, so we measured antioxidant-related enzymes. The concentration of MDA in the livers of the optimal concentration group was significantly reduced. These antioxidant indicators showed an increase in the antioxidant capacity of the liver. Oxidative stress can cause peroxidation of lipids, proteins, and other biological macromolecules, which can cause extensive damage to cells, including damage to cell structure and organ function [51,52]. Increase of antioxidant capacity represents the alleviation of oxidative stress. Shylaja et al. showed that eugenol has reactive oxygen scavenging activity, for example, it has a good scavenging effect on -OH and O_2_^−^ [53]. Eugenol could effectively reduce liver lipid peroxidation induced by thioacetamide and enhance liver antioxidant capacity in rats [54]. Previous studies have shown that the swimming speed of rainbow trout (*Oncorhynchus mykiss*) was reduced and response to external stimuli was delayed after treatment with eugenol [55]. This suggests that eugenol may also reduce stress in aquatic animals by inhibiting the central nervous system. 

The addition of an appropriate concentration of eugenol could also improve the antioxidant capacity of the gills, but could also increase the level of lipid peroxidation. The results showed that the relative expression of CAT was significantly up-regulated while the relative expression of Keap1 was significantly down-regulated, and the activity of GSH and the concentration of MDA were significantly increased when the eugenol concentrations were equal to or above 0.025 μL/mL. These results indicate that eugenol improves antioxidant capacity during transportation, but it does not prevent oxidative damage. This may be related to the fact that eugenol can negatively affect water quality. Fish gills are directly involved in the exchange of substances between the external environment and the organism’s interior, and gill tissue is the most sensitive to water pollutants [56]. Deterioration of water quality is usually accompanied by an increase in lipid peroxidation in the gills of aquatic animals. Previous studies have shown that ammonia exposure increased MDA concentrations in the gills of triangle sail mussels (*Hyriopsis cumingii*) and juvenile crucian carp (*Carassius auratus*) [57,58]. Thiobarbituric acid reactive substances (TBARS) in the gills of Brazilian flounder (*Paralichthys orbignyanus*) were significantly increased when the fish were exposed to ammonia nitrogen [59]. This suggests the necessity of feeding fish a diet containing immune stimulants before and after transportation, to alleviate lipid peroxidation in the gills during production practice. To prevent re-oxidative stress, a good water environment should be maintained after transportation. 

## 5. Conclusions

Our study shows that the addition of 0.05 μL/mL eugenol could maintain good water quality, reduce the metabolic capacities of liver and gills, and the antioxidant capacities of liver and gills, but could also increase levels of lipid peroxidation in the gills. We believe that our research may contribute to the green and healthy farming of the greater amberjack by reducing stress on the liver and gills during transport.

## Figures and Tables

**Figure 1 animals-12-02880-f001:**
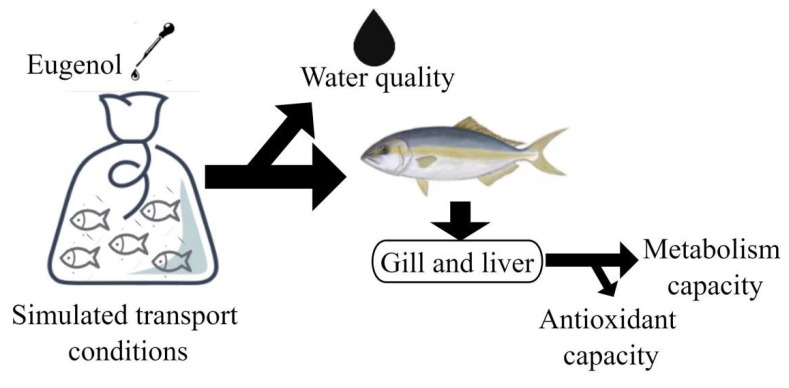
The schematic representation of the design and measured parameters for the simulated transport experiment.

**Figure 2 animals-12-02880-f002:**
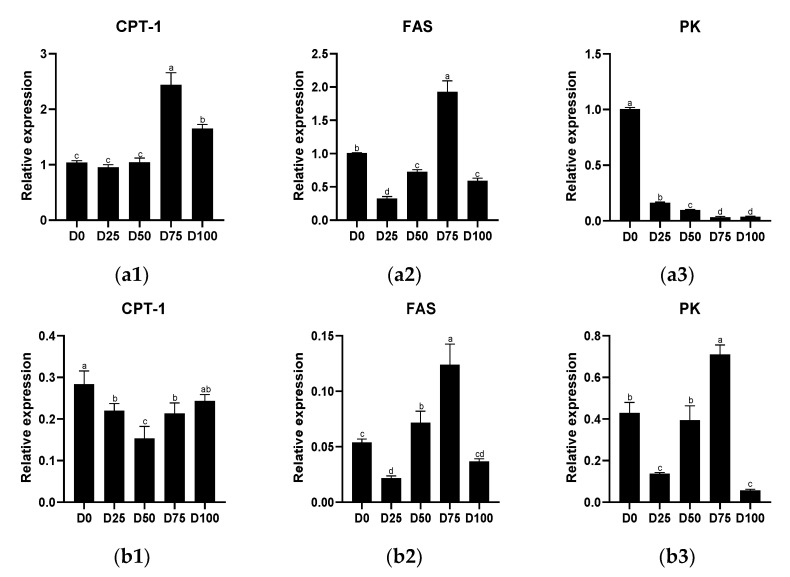
The effects of eugenol on the relative expression of CPT-1, FAS, PK in liver (**a1**–**a3**) and gill (**b1**–**b3**) tissue of greater amberjack under simulated transport conditions. Different letters above bars indicate significant differences at the 0.05 level.

**Figure 3 animals-12-02880-f003:**
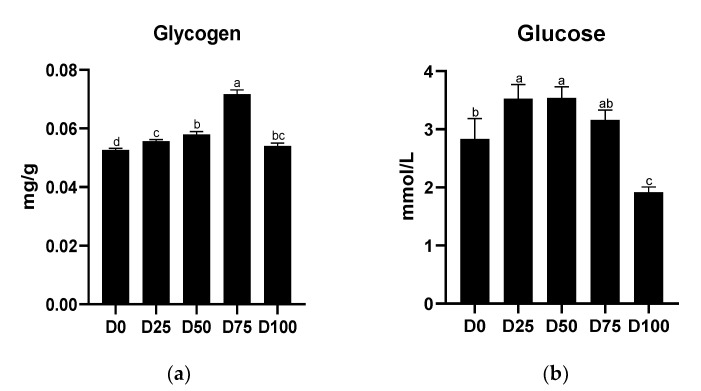
Effects of eugenol on the concentrations of (**a**) liver glycogen and (**b**) serum glucose in the greater amberjack under simulated transport conditions. Different letters above bars indicate significant differences at the 0.05 level.

**Figure 4 animals-12-02880-f004:**
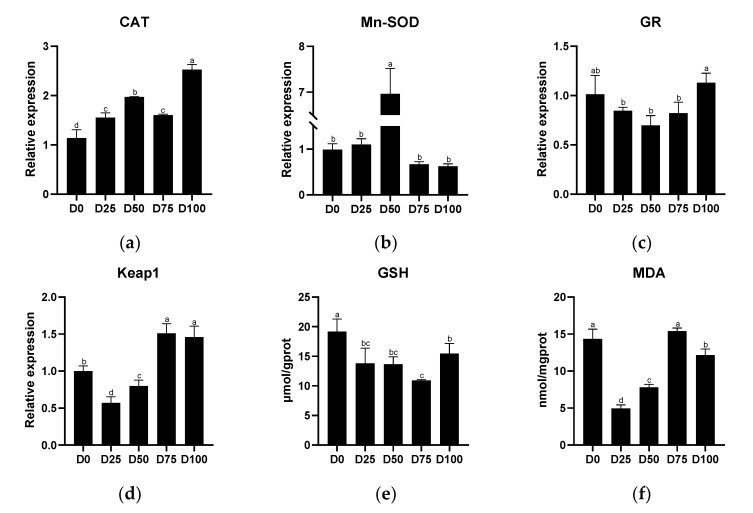
The effects of eugenol on antioxidant-related genes (**a**): CAT; (**b**): Mn-SOD; (**c**): GR; (**d**): Keap1, and antioxidant-related enzymes (**e**): GSH; (**f**): MDA, in the liver of greater amberjack under simulated transport conditions. Different letters above bars indicate significant differences at the 0.05 level.

**Figure 5 animals-12-02880-f005:**
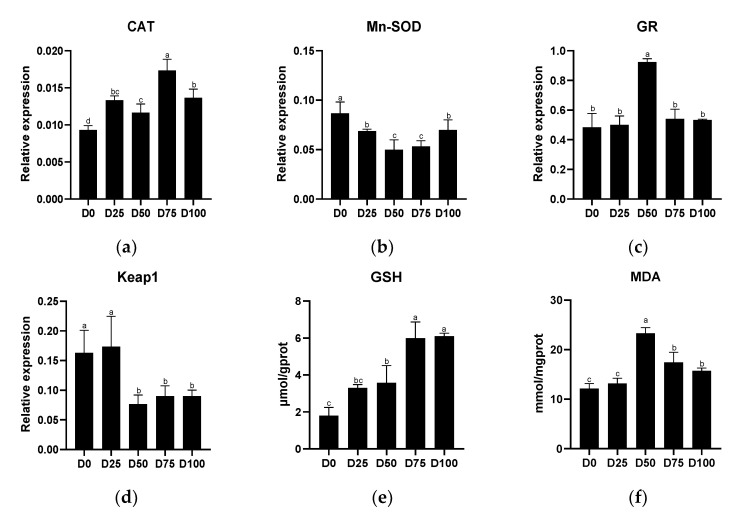
The effects of eugenol on antioxidant-related genes (**a**): CAT; (**b**): Mn-SOD; (**c**): GR; (**d**): Keap1, and antioxidant-related enzymes (**e**): GSH; (**f**): MDA, in gills of greater amberjack under simulated transport conditions. Different letters above bars indicate significant differences at the 0.05 level.

**Table 1 animals-12-02880-t001:** Primers used for real-time PCR.

Gene Abbreviation	Primer Sequence (5′-3′)	Amplicon Size (bp)	Accession Number
CPT-1 ^a^	F: GCAACACGGCAAGATGTCC	96	XM_022742255.1
	R: GAACCCTGGTAGCTGTAGAGTAGA		
FAS ^a^	F: GGCTATCTGTCGCACTTTCTG	173	XM_022765992.1
	R: ATTCACGCACTCGCTTCG		
PK ^a^	F: GATTGAGAATGGCGGTATGC	146	XM_022756239.1
	R: GGATGAAGGAGGCGAAGA		
CAT ^a^	F: CAAGTTTTACACTGAGGAGGGC	125	XM_022756059.1
	R: TGTGGGTTTGGGGATTGC		
Mn-SOD ^a^	F: CCAGCCTCAGCCAAACTAT	211	XM_022737832.1
	R: GCGGTCACATCTCCCTTT		
GR ^a^	F: TCACGAGCAGGAAGAGTCAG	106	XM_022760278.1
	R: TGCAGCATCTCATCACAGC		
Keap1	F: CCTCCATAAACCCACCAAAG	203	XM_022766859.1
	R: CAGCGTAGAAAAGCCCACT		
EF1-α ^b^	F: ATCGTTGCCGCTGGTGTT	134	XM_022744048.1
	R: TCGGTGGAGTCCATCTTGTT		
β-Actin ^b^	F: TCTGGTGGGGCAATGATCTTGATCTT	212	XM_022757055.1
	R: CCTTCCTTCCTCGGTATGGAGTCC		

^a^ Carnitine palmitoyltransferase-1 (CPT-1), fatty acid synthetase (FAS), pyruvate kinases (PK), catalase (CAT), manganese-superoxide dismutase (Mn-SOD) and glutathione reductase (GR). ^b^ EF1-α and β-Actin were used as reference genes.

**Table 2 animals-12-02880-t002:** Water quality parameters after transportation with different eugenol levels.

Indexes	Groups				
	D0	D25	D50	D75	D100
pH	6.58 ± 0.10 ^a^	6.74 ± 0.05 ^a^	6.70 ± 0.14 ^a^	6.80 ± 0.11 ^a^	6.78 ± 0.11 ^a^
DO (mg/L)	9.89 ± 0.16 ^b^	10.58 ± 0.19 ^b^	7.68 ± 0.24 ^c^	5.82 ± 0.10 ^d^	14.75 ± 0.24 ^a^
NH_4_^+^-N (mg/L)	0.91 ± 0.04 ^c^	0.85 ± 0.03 ^c^	1.30 ± 0.08 ^b^	2.13 ± 0.12 ^a^	0.92 ± 0.04 ^c^
NO_2_^−^-N (mg/L)	0.01 ± 0.00 ^c^	0.01 ± 0.00 ^c^	0.02 ± 0.00 ^b^	0.04 ± 0.00 ^a^	0.01 ± 0.00 ^c^

Notes: Eugenol concentration: 0 μL/mL (D0), 0.0125 μL/mL (D25), 0.025 μL/mL (D50), 0.0375 μL/mL (D75), 0.05 μL/mL (D100). Values with different superscript letters in the same row indicate differences (*p* < 0.05).

## Data Availability

The data that support the findings of this study are available from the corresponding author upon reasonable request.

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
