# Peer review of "Effects of Eugenol on Water Quality and the Metabolism and Antioxidant Capacity of Juvenile Greater Amberjack (Seriola dumerili) under Simulated Transport Conditions"

_animals, 2022, doi:10.3390/ani12202880_

Round 1
Reviewer 1 Report
Eugenol is one of the widely used fish anesthetic. The authors describe the influence of different doses of eugenol on some metabolic and antioxidant parameters in the gills and liver of greater amberjack (Seriola dumerili) under simulated transport conditions. The research describing the use of eugenol in great amberjack is scarce so the manuscript may further the knowledge regarding this fish species.
Although the topic is interesting the manuscript needs some major changes. The Table of results of water parameters that is described by the authors is not attached to the manuscript. Some of the cited articles concerns mammals or human and I would advise adding more articles concerning fish. More detailed comments are included in the attached file.

Author Response
We appreciate your patient suggestions to make us realize that there are so many errors in the manuscript, which we have corrected in the manuscript. Also, here is a reply to your question in the REVIEW.
1. Did you used the same equation as Cupp et al. to calculate the stock solution of eugenol? If yes, it would be helpful to include it in the manuscript.
Reply:We referenced the range of eugenol concentrations (0-30 mg/L) from cupp et al. And we used eugenol concentrations of 0-53.25 mg/L by unit conversion.
2. How long was the shaking simulating turbulence? Why the bags were shaken only 5 and not 8 times?
Reply: We sealed the bag containing the fish for 8 h, during which it was shaken to simulate the turbulence in transport, five times per hour, with each shaking lasting 1 min.
3. Missing water parameter results
Reply: We added it in the previous manuscript, but there may have been an error in the upload that caused this part of the results to not show up. We have put all the figures and tables at the end of the article.
4. Some of the cited articles concerns mammals or human and I would advise adding more articles concerning fish.
Reply: We have replaced some of the references with fish-related references.

Reviewer 2 Report
1. What is the main question addressed by the survey? R. Use of Eugenol in fish transport. 2. Do you consider the topic original or relevant in the area, and if then why? R. It is not an original topic, but it is relevant to the type of study, it is important to constantly evaluate the correct dosages. 3. What it adds to the subject area compared to other publications material? R. The research adds information for those who work with the species in question. 4. What improvements can the authors consider in relation to the method? R. Regarding the method, the statistics should consider regression analysis. 5. How are they consistent with how economies and address the main issue? R. Adequately, but the structure of the text does not allow the information to be interpreted properly. 6. Are referrals? R. Adequate. 7. Include any additional comments on the tables and figures. R. Adequate, but regression analysis can help to improve.
Long paragraphs that made continuous and objective reading difficult.
In statistics, the use of linear regression should be evaluated.
Author Response
Thank you for your suggestions.
- We have adjusted the discussion section of the manuscript to make the content more readable.
- We tried to perform linear regression analysis on the data, but we found that the R2 of the fitted linear regression equation was lower due to the large variation of some data, and we think it is not as good as the original result. The results of the linear regression equation have been added in the attachment, please check it and give us more suggestions.

Reviewer 3 Report
General comments:
The MS describes the use of different concentrations of euglenol in metabolism and antioxidant capacity in liver and gills of simulated transported conditions in the juvenile fish amberjack (Seriola dumerili). The MS was hard to review because line numbers were omitted along the text, please for further review line numbers should be added. The MS is quite interesting and is well written. However, there are several issues that should be cleared in order to improve the MS, particularly abstract, introduction and discussion section (see specific comments below). The MS should not be accepted as it is in the present form, therefore authors should be attend all comments and suggestions mentioned below and re submit the MS for a further reviewing.
Specific comments:
Abstract section
This should be improved. Results of gene expression are only mentioned. What is the interpretation of those results? Why some genes are up- and down regulated? What should be interpreted of differential gene expression between liver and gills?
Introduction section
Why the gills are considering as a main organ of fish metabolism?
Why the liver is considering a key organ in immune defense?
A brief description or definition of “stress” should be included. Why did you chose assess the gene expression of those enzymes? Which antioxidant mechanism does the fish have to counteract oxidative stress? What is oxidative stress? What is the relation of “stressful” transportation conditions with oxidative stress?
Results
Data in tables 1 and 2 are misplaced; indeed, there is one table with descriptions of primers used for gene expression. The table with data of water quality was omitted.
Star describing the results of glycogen in liver, and then glucose in serum. This in accordance as is showed in the figure 3.
Discussion section
The effects of different doses of euglenol on liver and gills should be discussed separately, since gene expression of enzymes assessed regulates different physiological aspects of fish. For instance, it is clear that enzymes in the liver regulate metabolism activity, but in the gills some other physiological aspects may be involved, probably related with anaerobic metabolism caused by decreased in the quality of water (e.g. decreased of dissolved oxygen) after maintained in the sealed plastic bags.
COR and GOT abbreviations should be properly described.
Water quality decreased by adding eugenol or by enclosed fish in the sealed plastic bags? This should be clearly described.
All the first paragraph of antioxidant capacity should be mentioned before in the introduction section. This is not discussion.
If euglenol is decreasing metabolism activity, why antioxidant capacity increased through gene and activity enzyme expression? Euglenol is acting as an antioxidant or pro-oxidant compound? It seems is actually acting as a compound that increased the potential of endogenous antioxidant mechanisms in the fish liver. This should be carefully discussed, and differences between gene and activity enzyme expression should be clearly stated. Please, also do not forget clearly discussed the physiological differences between the liver and gills.
Previously, was stated that D100 (0.05 ul/ml) treatment was considered the optimum concentration for fish transportation. Hereafter, at the end in the discussion section was mentioned that D25 (0.0125 ul/ml) was the optimum euglenol concentration for transportation of this fish species. What is the optimum concentration? This should be reviewed and properly clarified.
Author Response
Thank you very much for your advice!
1. This should be improved. Results of gene expression are only mentioned. What is the interpretation of those results? Why some genes are up- and down regulated? What should be interpreted of differential gene expression between liver and gills?
Reply: We have added the results of GSH, MDA, GLY,GLU and the interpretation of the results in the abstract. We believe that the gills, as an organ in direct contact with the water environment, are more sensitive to changes in the water environment and therefore the results for liver and gills are different.
2. Why the gills are considering as a main organ of fish metabolism?
Why the liver is considering a key organ in immune defense?
A brief description or definition of “stress” should be included. Why did you chose assess the gene expression of those enzymes? Which antioxidant mechanism does the fish have to counteract oxidative stress? What is oxidative stress? What is the relation of “stressful” transportation conditions with oxidative stress?
Reply: We replaced the reference on liver and gill function in lines 115-118. And we mentioned in lines 74-76 which factors are included in transport stress and added the section discussing part of the antioxidant capacity of aquatic animals in lines 76-89.
3. Data in tables 1 and 2 are misplaced; indeed, there is one table with descriptions of primers used for gene expression. The table with data of water quality was omitted.
Star describing the results of glycogen in liver, and then glucose in serum. This in accordance as is showed in the figure 3.
Reply: We've made changes to the table1 and table2 titles. We added water quality results to the previous manuscript, but there may have been an error in the upload that caused this part of the results to not show up. We have placed all the figures and tables at the end of the article.
4. The effects of different doses of euglenol on liver and gills should be discussed separately, since gene expression of enzymes assessed regulates different physiological aspects of fish. For instance, it is clear that enzymes in the liver regulate metabolism activity, but in the gills some other physiological aspects may be involved, probably related with anaerobic metabolism caused by decreased in the quality of water (e.g. decreased of dissolved oxygen) after maintained in the sealed plastic bags.
Reply: We have rewritten the discussion section as suggested.
5. (a)COR and GOT abbreviations should be properly described. (b) Water quality decreased by adding eugenol or by enclosed fish in the sealed plastic bags? This should be clearly described. (c) All the first paragraph of antioxidant capacity should be mentioned before in the introduction section. This is not discussion.
Reply: (a)The full names of COR and GOT have been added in line314. (b) this section has been adjusted in line296-302. (c)we have put this section in line76-89.
6. If euglenol is decreasing metabolism activity, why antioxidant capacity increased through gene and activity enzyme expression? Euglenol is acting as an antioxidant or pro-oxidant compound? It seems is actually acting as a compound that increased the potential of endogenous antioxidant mechanisms in the fish liver. This should be carefully discussed, and differences between gene and activity enzyme expression should be clearly stated. Please, also do not forget clearly discussed the physiological differences between the liver and gills.
Reply: We have revised this section as suggested in line335-341.
7. Previously, was stated that D100 (0.05 ul/ml) treatment was considered the optimum concentration for fish transportation. Hereafter, at the end in the discussion section was mentioned that D25 (0.0125 ul/ml) was the optimum euglenol concentration for transportation of this fish species. What is the optimum concentration? This should be reviewed and properly clarified.
Reply: The optimal concentration should be 0.05 ul/ml of eugenol when only water quality and metabolic results are considered. However, when the concentration of eugenol exceeds or equals to 0.025 ul/ml, the lipid peroxidation in the gills increases. Therefore, the optimal concentration of eugenol should be 0.0125 ul/ml when considering only the antioxidant capacity of liver and gills, and we have modified the conclusion section to conclude that the optimal concentration should be 0.05 ul/ml, because in practice it may be possible to mitigate lipid peroxidation in the gills by other means such as dietary supplementation with immune enhancers.

Round 2
Reviewer 1 Report
The manuscript has been improved considerably. I have some minor suggestions included in attached pdf file. There seems to be a problem with formatting after conversion from word to pdf in MDPI system.

Author Response
Thanks to your suggestions, we have revised the manuscript.

Reviewer 2 Report
Paper has been revised, by me it's ok.
Author Response
Thank you for your review!
Reviewer 3 Report
Almost all comments and suggestion were properly attended, there are only some comments in pages 7 to 9 that should be attended before the acceptance of this MS for been published in this journal.

Author Response
Thanks to your suggestions, we have made changes to pages 7-9. In addition, the tables and figures were placed after the references in the manuscript we provided, and there may have been a problem converting word to pdf in the review system.
